# Erosion Corrosion Behavior of Aluminum in Flowing Deionized Water at Various Temperatures

**DOI:** 10.3390/ma13030779

**Published:** 2020-02-08

**Authors:** Liangshou Hao, Feng Zheng, Xiaoping Chen, Jiayang Li, Shengping Wang, Youping Fan

**Affiliations:** 1Tianshengqiao Bureau, Extra High Voltage Power Transmission Company, China Southern Power Grid (CSG), Xingyi 562400, China; haoliangshou@im.ehv.csg (L.H.); zhengfeng@im.ehv.csg (F.Z.); chenxiaoping@im.ehv.csg (X.C.); lijiayang@im.ehv.csg (J.L.); 2Faculty of Material Science and Chemistry, China University of Geosciences, Wuhan 430074, China; 3School of Electrical Engineering and Automation, Wuhan University, Wuhan 430072, China

**Keywords:** molar activation energy, equilibrium constant, aluminum, erosion corrosion, deionized water, radiator, high voltage direct current

## Abstract

To optimize the operating temperature and flow velocity of cooling water in a high voltage direct current (HVDC) thyristor valve cooling system, the erosion corrosion characteristics of aluminum electrodes in deionized water at various temperatures were studied. With increasing water temperature, the corrosion current of the aluminum electrode gradually increases and the charge transfer impedance gradually decreases, thus, the corrosion of aluminum tends to become serious. The aluminum electrode in 50 °C deionized water has the most negative corrosion potential (−0.930 V), the maximum corrosion current (1.115 × 10^−6^ A cm^−2^) and the minimum charge transfer impedance (8.828 × 10^−6^ Ω), thus, the aluminum corrosion at this temperature is the most serious. When the temperature of deionized water increases, the thermodynamic activity of the ions and dissolved oxygen in the deionized water increases, and the mass transfer process accelerates. Therefore, the electrochemical corrosion reaction of the aluminum surface will be accelerated. The corrosion products covering the aluminum electrode surface are mainly Al(OH)_3_. With increasing water temperature, the number of pits and grooves formed by corrosion on the aluminum surface increased. In this paper, the molar activation energy Ea and the equilibrium constant *K* of the aluminum corrosion reaction with various temperatures are calculated. This clarifies the effect of temperature on the aluminum corrosion reaction, which provides a basis for protecting aluminum from corrosion. The results of this study will contribute to research that is focused on the improvement of production techniques used for HVDC thyristor valve cooling systems.

## 1. Introduction

When a high voltage direct current (HVDC) converter works, a large amount of heat is transferred to an aluminum radiator that is in direct contact with the thyristor. Then, the heat is conducted out of the system by the cooling water flowing in the inner hole of the aluminum radiator to maintain the normal operating temperature of the converter. The cooling water is deionized water with very low conductivity. The scaling that forms on grading electrodes in HVDC cooling circuits that use deionized water is a long-known and unsolved problem and has a great impact on the safe operation of an HVDC cooling system [1,2]. The precipitates on the grading electrode can be aluminum oxide or hydroxide. The primary cause of scaling on a platinum grading electrode scaling is corrosion of the aluminum radiator [3,4]. In other electrochemical systems, some researchers had studied the corrosion behavior in high conductivity NaCl solutions [5,6] and applied organic corrosion inhibitors to suppress aluminum corrosion [7,8]. However, considering that deionized water needs to ensure low conductivity, these research are not practical effect in HVDC systems. Therefore, it is necessary to study the corrosion characteristics of aluminum under a similar environment that uses internal cooling water in an HVDC system. To reduce the corrosion of aluminum in HVDC thyristor valve cooling systems, the corrosion behaviors of aluminum immersed in deionized water, ammonia [9], carbon dioxide [10] and sodium bicarbonate [11] at a specific temperature have been reported previously. The temperature and flow velocity of the cooling water not only controls the cooling effect but also affects the corrosion rate of the inner aluminum radiator surface and the scaling rate of the platinum grading electrode. The temperature of the deionized water in direct contact with the interior is an important factor affecting the electrochemical corrosion of aluminum [12]. The water temperature directly affects the kinetics constant of the mass transfer process. The corrosion rate of the metal doubles, which is controlled by the charge transfer process with a temperature increase of 10 K or by a diffusion process for a temperature increase of 30 K [13]. The temperature can also change the corrosion of aluminum and the scaling of the platinum grading electrode by affecting the solubility of the reactants and products [14]. In addition, with increasing temperature, the solubility of oxygen decreases, and the biological activity increases, which affects the corrosion rate of metals [15].

It is necessary to clarify the corrosion characteristics of aluminum in deionized water at various temperatures. Under the premise of ensuring the cooling effect of the HVDC valve cooling system, the conclusion can guide the optimization of the temperature and flow velocity of cooling water in an HVDC valve cooling system; the above will be helpful for improving production efficiency. In this paper, a pipe-flow experiment method is used to simulate an erosion corrosion experiment with aluminum electrodes, and the erosion corrosion behavior of aluminum in deionized water with various temperatures is discussed.

## 2. Materials and Methods 

### 2.1. Erosion Corrosion Experiment

A pipe flow experiment method was used for testing. The erosion corrosion experimental device is shown in Figure 1. The working electrode, the counter electrode and the reference electrode were fixed in a Teflon pipe a diameter of 10 cm to simulate the erosion corrosion process in an aluminum radiator. The temperatures of the deionized waters were 10, 20, 30, 40, and 50 °C. A rotameter was used to control the flow rate of the deionized water in the pipe, and the water flow rate in all tests was 3 m s^−1^.

### 2.2. Electrochemical System

The aluminum work electrodes were cut from an HVDC aluminum radiator, which was composed of Si (0.57 wt%), Fe (0.63 wt%), Cu (0.14 wt%), Mn (1.27 wt%), Zn (0.09 wt%), Li (0.03 wt%), and Al (97.31 wt%) [16]. An electrode work surface of 1 cm^2^ was retained, and the rest was coated with epoxy resin. Before testing, the working electrodes were polished with diamond paper and nanoalumina powder and cleaned with deionized water and absolute ethanol.

A platinum black electrode was used as the counter electrode, and the reference electrode was a saturated calomel electrode (SCE). The potentials of the SCE at 10, 20, 30, 40, and 50 °C were 0.254, 0.247, 0.241, 0.234, and 0.228 V (relative to the standard hydrogen electrode (SHE)), respectively. The test electrolytes were deionized water, and their conductivities at 10, 20, 30, 40, and 50 °C were approximately 0.139, 0.141, 0.145, 0.147, and 0.150 μS cm^−1^, respectively. The new deionized water was used in each test.

### 2.3. Electrochemical Test

Tafel polarization curves and electrochemical impedance spectroscopy (EIS) spectra were obtained using a CHI660D electrochemical workstation (CHI, Shanghai, China). The Tafel polarization curves were tested at the open circuit potential, and the potential scan rate was 1 mV s^−1^. The potential range was 0.8 V (ranging from the potential 0.4 V lower than the stable potential to the potential 0.4 V higher than the stable potential). The corrosion potentials and corrosion current densities were obtained from the Tafel polarization curves. The corrosion characteristics of the aluminum electrode surfaces were determined from the results of EIS testing at the open circuit potential. The frequency range of the EIS was from 1 Hz to 10^5^ Hz with an amplitude of 5 mV.

The aluminum working electrodes were eroded for 1, 3, 5, 7, and 9 h, and then the Tafel polarization curves and EIS tests were conducted. The stable potential was recorded for each electrochemical test.

### 2.4. Characterization

The electrochemical corrosion rate of aluminum in deionized water was slow. To more quickly obtain the corrosion surface and corrosion products of aluminum, the samples for scanning electron microscopy (SEM), energy dispersive spectroscopy (EDS), Fourier transform infrared (FTIR) and powder X-ray diffraction (XRD) were subjected to accelerated corrosion by potentiostatic anodic oxidation at 0.5 V (vs. Pt) for 0.5 h [17,18]. Large area platinum electrodes were used as the cathodes, 1 cm × 1 cm aluminum foils after polishing and washing were used as the anode, and the flowing deionized water with 3 m s^-1^ was used as the electrolyte at 10, 20, 30, 40, and 50 °C.

The corrosion product compositions were determined using a D8-Focus X-ray powder diffraction instrument with a Cu target (Bruker, Karlsruhe, Germany). The scanning angle range was from 5 to 80 degrees, and the scan rate was 8° min^−1^. The chemical bonds of the corrosion products were characterized using a Nicolet 6700 Fourier transform infrared spectrometer (Thermo Fisher, Waltham, USA). SEM images were obtained using an SU8010 ultrahigh-resolution field emission scanning electron microscope capable of high-performance X-ray energy dispersive spectroscopy (Hitachi, Tokyo, Japan).

## 3. Results and Discussion

### 3.1. Tafel Polarization Curves

Tafel polarization curves of aluminum electrodes with various temperatures are shown in Figure 2. The corrosion potential of aluminum gradually and negatively shifted with increasing temperature, which indicated that the corrosion rate of aluminum accelerated. Moreover, with the increase in erosion time, the corrosion potential of aluminum at a temperature gradually and negatively shifted, indicating that the corrosion tendency of aluminum increased.

The corrosion potentials and corrosion current densities of the aluminum electrode with various temperatures are shown in Table 1. With increasing erosion time, the corrosion current of aluminum with various temperatures decreased first and then increased. During the initial stage of corrosion, aluminum corrosion was suppressed because oxide films were formed on the aluminum surfaces as protective layers, but then the corrosion of the aluminum electrode surfaces continued to occur. The corrosion current reached a minimum at 5 h when the temperature was 10, 20, and 30 °C, and the corrosion current reached a minimum at 3 h when the temperature was 40 and 50 °C. The above results indicated that the formation of oxide films on the aluminum surfaces was faster at high temperatures than at low temperatures. In addition, the corrosion current increased significantly with increasing temperature. The aluminum corrosion increased with increasing temperature. The corrosion and dissolution of aluminum occurred during the anodic polarization curve [19]. The aluminum electrodes with the same potential change had small anodic Tafel slopes, fast corrosion current changes, and fast corrosion reaction rates. The anodic Tafel slopes of aluminum at various temperatures are shown in Table 1. With an increase in water temperature, the anodic Tafel slope of aluminum gradually decreased and the change rate of the aluminum corrosion current increased, thus, the aluminum corrosion became more serious.

With an increase in water temperature, the thermodynamic activity of the ions and dissolved oxygen in the water increased [20], and the mass transfer processes were accelerated, which accelerated the reaction rate of Al^3+^ and OH^−^ in the electrolyte and promoted the anodic and cathodic reactions. The above was the reason why the aluminum corrosion became more serious.

### 3.2. EIS

The EIS curves and the associated equivalent circuit diagram of the aluminum electrodes at various temperatures are shown in Figure 3. The Nyquist diagrams of the aluminum corrosion at various temperatures were similar, with a semicircle and a straight line. The semicircular diameter in the high frequency region represents the charge transfer impedance during the corrosion process, which reflects the corrosion resistance of the aluminum in the solution. For the equivalent circuit diagram of the EIS (Figure 3f), R1 represented the solution resistance between the aluminum electrode and the reference electrode, R2 represented the impedance of the electrolyte through the deposition layer, R3 represented the charge transfer impedance for oxidation of aluminum, C1 represented the capacitance of the cladding layer, C2 represented the capacitance of the double layer, and W represented the diffusion impedance of ions in the electrolyte [21,22]. The fitted curves matched well to the experimental data, which indicated that the equivalent circuit diagram was representative of the corrosion reaction of aluminum in the deionized water at different temperatures.

The data for the corresponding numerical simulation of the equivalent circuit are shown in Table 2. With the increase in the water temperature, the charge transfer impedance gradually decreased, indicating that the corrosion resistance of aluminum in the deionized water gradually decreased, and that the aluminum corrosion intensified. At the same temperature, the charge transfer resistance of aluminum first increased and then decreased with increasing erosion time. The above result was because a protective oxide film layer was formed on the aluminum surface at the initial stage of corrosion to suppress aluminum corrosion, but after an extended period of time, corrosion of the aluminum surface would continue to occur. The results were consistent with the polarization curves. A high water temperature could aggravate aluminum corrosion.

### 3.3. SEM

The surface morphologies of the aluminum electrodes after being corroded with potentiostatic anodic oxidation at 0.5 V for 0.5 h in deionized water at various temperatures are shown in Figure 4. Obvious corrosion phenomena could be observed on the corroded aluminum surface when the water temperatures were 40 (Figure 4d) and 50 °C (Figure 4e). There were gullies, pits, and corrosion products. In contrast, aluminum corrosion in low-temperature deionized water was relatively mild. As shown in Figure 4a–c, only a small number of corrosion pits and gullies appeared on the electrode surface. With an increase in water temperature, the corrosion pits became larger and the corrosion products were more abundant, thus, the aluminum surface corrosion was more serious. These results were consistent with the Tafel polarization curves and EIS.

The SEM image for the cross-section of the aluminum electrode after potentiostatic anodic oxidation at 0.5 V for 0.5 h in 50 °C deionized water is shown in Figure 4f. There were significant differences between the corrosion product layer on the outside surface and the uniform and dense aluminum interior. The thickness of the corrosion product layer was ~40 µm.

### 3.4. EDS

The SEM image and EDS diagrams of the corrosion products on the aluminum electrode surface after potentiostatic anodic oxidation at 0.5 V for 0.5 h in 50 °C deionized water are shown in Figure 5. The elemental compositions of the corrosion products were mainly aluminum and oxygen. According to the EDS elemental diagram, the atomic ratio of aluminum and oxygen was 84.2:15.8. Aluminum and oxygen were uniformly distributed on the electrode surface. Aluminum might come from corrosion layer and substrate, while oxygen was mainly concentrated in the corrosion products. Since the corrosion of the aluminum electrode surface was more serious at high temperature, there were more corrosion products on the aluminum surface at high temperatures. At this time, it could be inferred that the corrosion products on the aluminum surface were Al(OH)_3_ or Al_2_O_3_. The corrosion layers were relatively compact and had strong adhesion with the base metal. This might inhibit the deeper corrosion in the later stage.

### 3.5. IR and XRD

The corrosion products on the aluminum electrode surfaces were tested by IR and XRD. The IR spectrum of the corrosion products is shown in Figure 6a. The sample had a series of strong absorption peaks at wavenumbers of 3286, 2967, 2924, and 2861 cm^−1^, which were caused by the stretching vibration of hydroxyl (-OH). The absorption peaks at wavenumbers of 1533 and 1452 cm^−1^ were caused by the deformation vibration of hydroxyl (-OH). The stretching vibrations of the absorption peaks at 1067, 1003 and 910 cm^−1^ corresponded to the Al-O bond [23,24]. The IR results confirmed that the corrosion products were mainly Al(OH)_3_.

The XRD spectra of the corrosion products on the aluminum electrode surface after potentiostatic anodic oxidation at 0.5 V for 0.5 h in deionized water at various temperatures are shown in Figure 6b. For all the samples, four strong peaks were observed at 39°, 45°, 65°, and 78°, which matched well to Al (PDF #04-0708) [25]. The peaks at 25°, 57°, and 63° could also correspond to the peaks in the standard card for Al(OH)_3_ (PDF #26-0025) [26]. Thus, the corrosion products were mainly Al(OH)_3_. The aluminum corrosion in deionized water accelerated with a constant potential anodic oxidation. The hydrogen evolution reaction at the cathode made the solution alkaline, and the anode dissolution reaction and corrosion reaction occurred at the anode. The corrosion products in the alkaline environment were basically Al(OH)_3_. In addition, the corrosion time was short, and Al(OH)_3_ could not be further converted into Al_2_O_3_. With increasing water temperature, the peaks of the Al(OH)_3_ corrosion product at 25°, 57°, and 63° gradually increased. As the temperature increased, the amount of corrosion products on the aluminum surface gradually increased, and the aluminum corrosion gradually became serious.

### 3.6. Thermodynamics of the Aluminum Corrosion Process

The previous results from the electrochemical tests show that increasing water temperature could aggravate aluminum corrosion. It was observed that with an increased water temperature, the thermodynamic activity of the ions and dissolved oxygen in the deionized water increased [20], and the mass transfer process was accelerated, which accelerated the reaction rate of Al^3+^ and OH^−^ in the deionized water and promoted the anodic and cathodic reactions. The above was the reason why the electrochemical reaction on the aluminum surface intensified, and the aluminum corrosion became more serious. In addition, the above observations explain the exacerbation of aluminum corrosion from the experimental dynamics. The following attempts were made to explore the thermodynamic characteristics of aluminum corrosion reactions.

Based on the Arrhenius formula [27], the molar activation energy *E*a of the aluminum corrosion reaction could be calculated from the self-corrosion current density *i*_corr_:(1)$$K=K_{0}e^{-\frac{E_{a}}{RT}}$$
(2)$$\ln i_{\mathrm{corr}}=\ln A-\frac{E_{a}}{RT}$$
where *i*_corr_ is the self-corrosion current density, *E*_a_ is the molar activation energy, A is the pre-exponential factor, R is the gas constant (8.314 mol^−1^ K^−1^), and *T* is the absolute temperature. Plotting the self-corrosion current density *i*_corr_ at each temperature when the erosion corrosion time was 9 h, the Arrhenius linear fitting curve is shown in Figure 6c. The molar activation energy of the aluminum corrosion reaction was 25.956 kJ mol^−1^, which could be obtained from the linear slopes of ln*i*_corr_ and 1/*T*.

EDS, IR, and XRD characterization confirmed that the corrosion products were mainly Al(OH)_3_. The corrosion reaction of the aluminum anode can be written as the following reaction (Equation (3)):(3)$${\mathrm{Al}+3\mathrm{OH}}^{-}{=\mathrm{Al}(\mathrm{OH})}_{3}{+3e}^{-}$$

Manipulating Equations (1) and (2), they can be transformed into Equation (4):(4)$$\ln\frac{K_{2}}{K_{1}}=\frac{E_{a}}{R}\left(\frac{1}{T_{1}}-\frac{1}{T_{2}}\right)$$
(5)$$\Delta_{r}G_{m}^{\theta }=-RT\ln K^{\theta }$$

Based on the data in Lange’s Handbook of Chemistry [28], the Gibbs free energy of the aluminum corrosion reaction (Equation (3)) under standard conditions (100 kPa, 298.15 K) could be calculated, and $\Delta_{r}G_{m}^{\theta }$ was 159.5 kJ mol^−1^. According to Equation (5), the corrosion reaction rate of aluminum could be calculated under standard conditions, and *K*^θ^ was 1.136 × 10^−28^. According to Equation (4), the corrosion reaction rate K of aluminum could be obtained as shown in Table 3.

As shown in Table 3, with an increase in water temperature, the corrosion reaction equilibrium constant K continuously increased, the corrosion reaction rate continuously accelerated, and the aluminum corrosion gradually became more serious. When the temperature increased, the activation energy *E*a required for the corrosion reaction hardly changed, but more reactive ions H^+^, OH^−^, Al^3+^ in the deionized water reached "activation" (the thermodynamic activity increased), thus, more ions could participate in the corrosion reaction. In addition, the enthalpy change of the aluminum corrosion reaction under standard conditions (100 kPa, 298.15 K) could be calculated, $\Delta_{r}H_{m}^{\theta }$ was 383 kJ mol^−1^, and greater than 0. Therefore, the aluminum corrosion reaction was an endothermic reaction. According to Le Chatelier’s principle, with an increase in water temperature, the endothermic reaction would proceed in the forward direction, and the corrosion reaction rate would accelerate. The thermodynamic calculation results here were also consistent with the previous electrochemical test and physical characterization results.

## 4. Conclusions

Erosion corrosion characteristics of aluminum electrodes in flowing deionized water with 3 m s^−1^ at various temperatures (10, 20, 30, 40, and 50 °C) were given in this paper. The corrosion products that covered the surface of the aluminum electrode were mainly Al(OH)_3_. With increasing water temperature, the corrosion current of aluminum gradually increased, the charge transfer impedance gradually decreased, and the corrosion of aluminum became more serious. The aluminum electrode in 50 °C deionized water has the most negative corrosion potential (−0.930 V), the maximum corrosion current (1.115 × 10^−6^ A cm^−2^), and the minimum charge transfer impedance (8.828 × 10^−6^ Ω). The aluminum corrosion at this temperature was the most serious, and the reasons were the thermodynamic activity of the ions and dissolved oxygen in the deionized water increased. The activation energy and the equilibrium constant of the aluminum corrosion reaction was calculated based on the self-corrosion current density, and the mechanism of the effect of temperature on the corrosion reaction of aluminum was clarified from thermodynamic calculations. The high temperature of deionized water was not conducive to the corrosion protection of aluminum.

This work focused on actual production, and the results provide a scientific basis for corrosion protection of the applied aluminum radiators in HVDC thyristor valve cooling systems. Reducing the temperature of deionized water in inner cooling systems could slow down the corrosion of the aluminum radiator and reduce the flow rate of water. However, that would increase the workload of the external cooling system.

## Figures and Tables

**Figure 1 materials-13-00779-f001:**
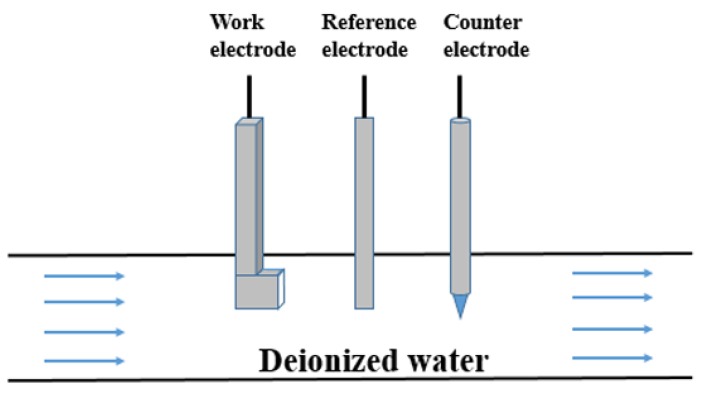
Diagram of the experimental device for the erosion corrosion experiment.

**Figure 2 materials-13-00779-f002:**
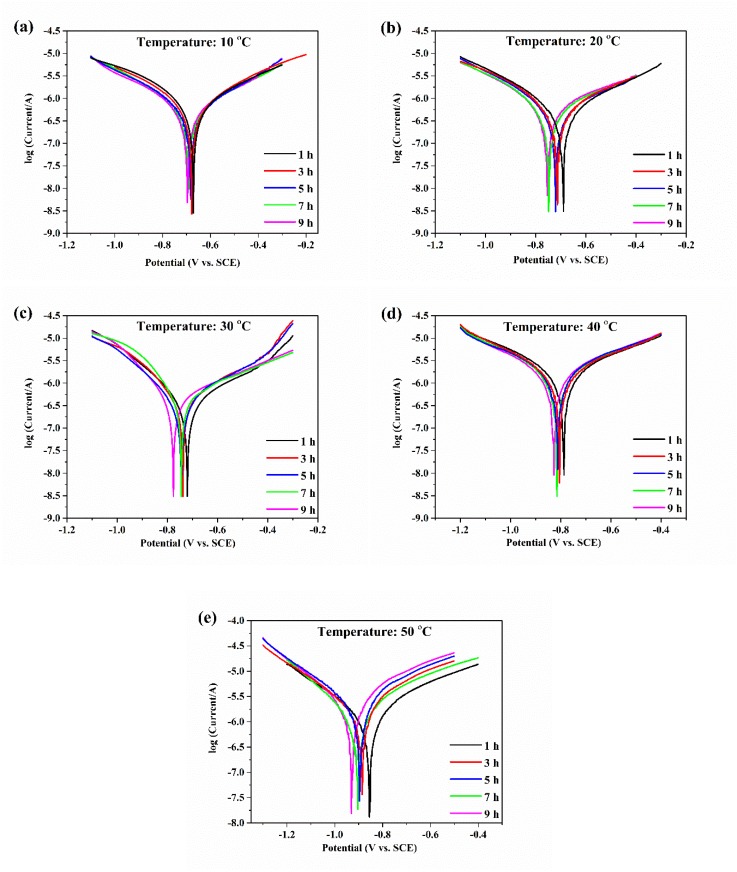
The Tafel polarization curves of aluminum electrodes in deionized water at 10 (**a**), 20 (**b**), 30 (**c**), 40 (**d**), and 50 °C (**e**).

**Figure 3 materials-13-00779-f003:**
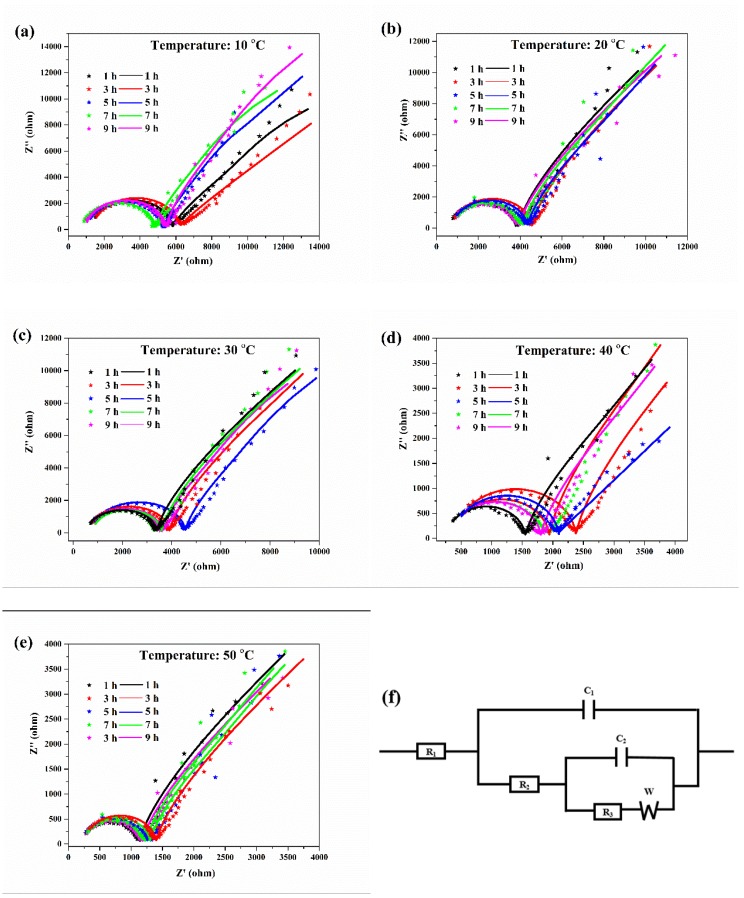
EIS curves for aluminum electrodes in deionized water at 10 (**a**), 20 (**b**), 30 (**c**), 40 (**d**), 50 °C (**e**), and the associated equivalent circuit diagram (**f**). The original data and fitting curves are indicated by the dotted lines and the solid lines, respectively.

**Figure 4 materials-13-00779-f004:**
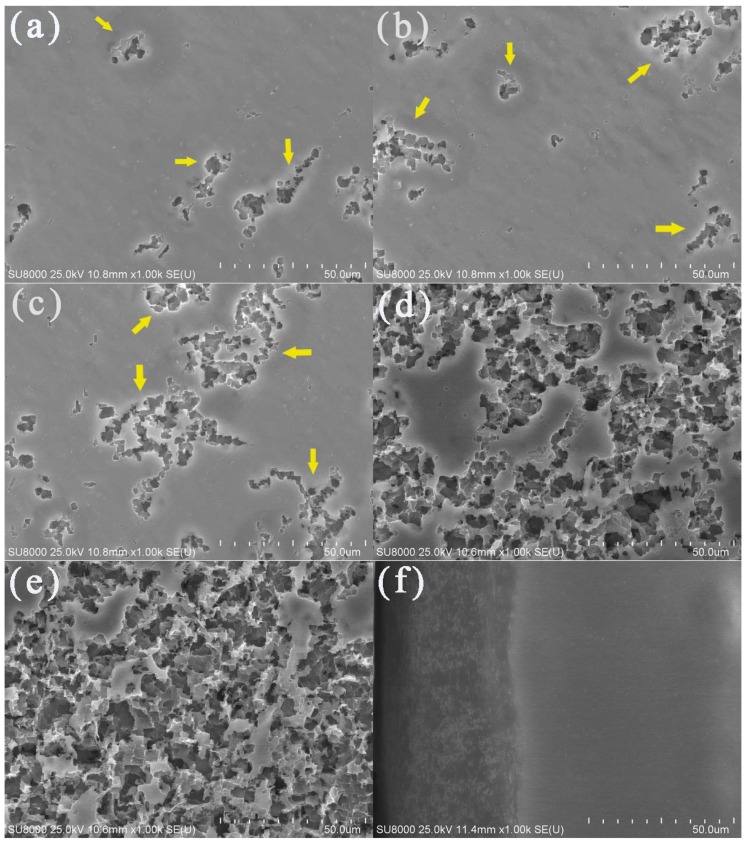
SEM images of the aluminum electrode surfaces in deionized water at 10 (**a**), 20 (**b**), 30 (**c**), 40 (**d**), and 50 °C (**e**) and the cross-section of the aluminum electrode after a constant voltage corrosion of 0.5 V for 0.5 h in 50 °C deionized water (**f**). The yellow arrows referred to the gullies and pits.

**Figure 5 materials-13-00779-f005:**
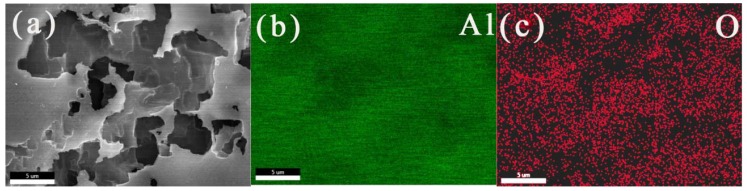
SEM images (**a**) and the Al (**b**) and O (**c**) EDS elemental analyses for the corrosion products on the aluminum electrode surface after a constant voltage corrosion of 0.5 V for 0.5 h in 50 °C deionized water.

**Figure 6 materials-13-00779-f006:**
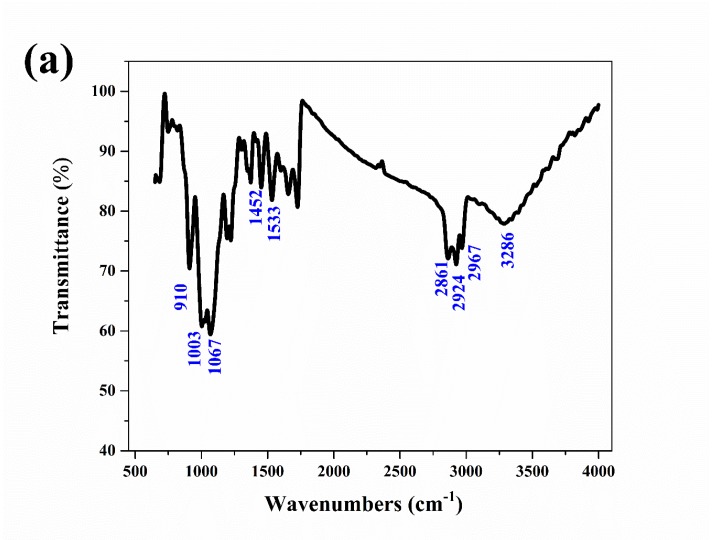
IR spectrum (**a**) of the aluminum electrode after a constant voltage corrosion of 0.5 V for 0.5 h in 50 °C deionized water, XRD spectra (**b**) of the aluminum electrode after a constant voltage corrosion of 0.5 V for 0.5 h at various temperatures and the Arrhenius linear fitting curves (**c**) of aluminum at various temperatures.

**Table 1 materials-13-00779-t001:** Corrosion potentials and current densities of aluminum electrodes in deionized water at various temperatures.

Temperature (°C)	Flushing Time (h)	Corrosion Potential (V)	Corrosion Current Density (10^−7^ A cm^-2^)	Cathodic Tafel Slope (V dec^−1^)	Anodic Tafel Slope (V dec^−1^)
10	1	−0.671	2.733	−0.183	0.223
	3	−0.678	2.646	−0.185	0.222
	5	−0.682	2.302	−0.177	0.229
	7	−0.693	2.495	−0.183	0.220
	9	−0.707	2.903	−0.193	0.228
20	1	−0.696	3.091	−0.194	0.222
	3	−0.713	2.647	−0.168	0.214
	5	−0.715	2.553	−0.179	0.219
	7	−0.748	2.800	−0.183	0.218
	9	−0.751	3.646	−0.177	0.223
30	1	−0.721	4.897	−0.187	0.216
	3	−0.738	3.971	−0.191	0.214
	5	−0.744	3.878	−0.184	0.216
	7	−0.745	4.522	−0.173	0.213
	9	−0.776	4.166	−0.172	0.215
40	1	−0.787	9.085	−0.197	0.211
	3	−0.805	8.064	−0.193	0.214
	5	−0.812	8.424	−0.200	0.205
	7	−0.815	8.552	−0.199	0.211
	9	−0.827	8.510	−0.196	0.214
50	1	−0.855	11.15	−0.199	0.202
	3	−0.891	9.634	−0.178	0.209
	5	−0.897	9.922	−0.187	0.205
	7	−0.904	9.959	−0.175	0.204
	9	−0.930	10.60	−0.179	0.205

**Table 2 materials-13-00779-t002:** EIS parameters of aluminum electrodes obtained by fitting the data to the equivalent circuit model.

Temperature (°C)	Flushing Time (h)	R1 (Ω)	R2 (Ω)	R3 (10^−6^ Ω)	C1 (10^−9^ F)	C2 (10^−6^ F)
10	1	346.2	3985	53.475	7.263	5.634
	3	329.4	4623	63.145	7.132	5.551
	5	356.7	4514	60.548	6.523	5.267
	7	326.8	4238	58.423	7.032	5.589
	9	315.4	4176	57.872	7.462	5.647
20	1	276.1	3551	48.624	7.378	5.526
	3	264.3	4021	53.684	7.034	5.301
	5	284.9	3967	52.167	6.047	5.182
	7	215.7	3826	50.872	7.099	5.459
	9	264.8	3762	50.076	7.367	5.498
30	1	422.8	2783	36.094	7.338	5.613
	3	454.3	3411	49.460	6.860	5.357
	5	463.1	3955	51.754	6.526	5.154
	7	513.5	3153	42.542	6.921	5.375
	9	435.1	3013	38.164	6.389	5.986
40	1	340.5	1462	17.53	3.458	4.887
	3	274	2284	32.237	5.287	4.981
	5	285.7	1715	28.046	4.027	6.773
	7	279.7	1564	21.377	3.407	5.440
	9	302.8	2836	19.159	4.754	3.849
50	1	236.4	899.8	8.828	9.009	6.594
	3	201.5	990	17.513	5.224	7.513
	5	233.3	968	15.388	7.098	5.605
	7	249.1	983.6	13.714	8.028	6.084
	9	242.4	939.3	11.689	8.642	6.227

**Table 3 materials-13-00779-t003:** Gibbs free energies and corrosion reaction equilibrium constants of aluminum at various temperatures.

Temperature (°C)	10	20	30	40	50
$\Delta_{r}G_{m}$ (kJ mol^−1^)	152.78	157.26	161.74	166.22	170.70
K	6.523 × 10^−29^	9.502 × 10^−29^	1.350 × 10^−28^	1.876 × 10^−28^	2.554 × 10^−28^

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
