# Peer review of "Erosion Corrosion Behavior of Aluminum in Flowing Deionized Water at Various Temperatures"

_materials, 2020, doi:10.3390/ma13030779_

Round 1

Reviewer 1 Report

It is an interesting work, well organized and presented. Minor spelling corrections would improve the quality of the article, which is complementary to ref [7] by the same authors. 

Reviewer 2 Report

The paper is logically structured and well understandable.

When describing the test setup, it should be mentioned the flow rate and whether the medium was running in loop or straight (i.e. always new solution).

Page 9, line 245 and 246: Literature reference to datasheets PDF #040-708 and #26-0025 ais missing. What catalogue was used?

Everything else seems clear to me.

Reviewer 3 Report

Review for materials- 705700

Erosion corrosion behavior of aluminum in flowing deionized water at various temperatures

The authors present an interesting research work for the readers of the journal Materials. Anyway, I present several recommendations:

* For the sake of clarity, please improve the quality of figure resolution (e.g. Figure 2 need a larger font size in the axis text, Figure 6 should be placed vertically so that each graphic is better visualized, etc.)

* Please, explain better the figure 4: authors could include arrows indicating the location of pits and gullies in the images.

* In my opinion, Figure 5 must be accompanied by an expanded explanation.

* Authors should be improved the Conclusion section to give more relevance to the paper.

* It would be advisable to include some papers from the journals of MDPI editorial (Metals, Applied Sciences, Materials, etc.) related to the topic of the manuscript.

Round 2

Reviewer 3 Report

Accept in present form.